# The Landscape of Aberrant Alternative Splicing Events in Steatotic Liver Graft Post Transplantation via Transcriptome-Wide Analysis

**DOI:** 10.3390/ijms24098216

**Published:** 2023-05-04

**Authors:** Hui Liu, Yueqin Zhu, Kevin Tak-Pan Ng, Chung-Mau Lo, Kwan Man

**Affiliations:** 1Department of Surgery, School of Clinical Medicine, HKU-SZH & LKS Faculty of Medicine, The University of Hong Kong, Hong Kong, China; 2Department of Pathophysiology, Key Laboratory of Cell Differentiation and Apoptosis of the Chinese Ministry of Education, Shanghai Jiao Tong University School of Medicine, Shanghai 200025, China

**Keywords:** liver transplantation using steatotic graft, tumor recurrence, alternative splicing events, metabolism and immune cells, cancer hallmarks

## Abstract

The application of steatotic liver graft has been increased significantly due to the severe donor shortage and prevalence of non-alcoholic fatty liver disease. However, steatotic donor livers are vulnerable to acute phase inflammatory injury, which may result in cancer recurrence. Alternative splicing events (ASEs) are critical for diverse transcriptional variants in hepatocellular carcinoma (HCC). Here, we aimed to depict the landscape of ASEs, as well as to identify the differential ASEs in steatotic liver graft and their association with tumor recurrence after transplantation. The overall portrait of intragraft transcripts and ASEs were elucidated through RNA sequencing with the liver graft biopsies from patients and rat transplant models. Various differential ASEs were identified in steatotic liver grafts. CYP2E1, ADH1A, CYP2C8, ADH1C, and HGD, as corresponding genes to the common pathways involved differential ASEs in human and rats, were significantly associated with HCC patients’ survival. The differential ASEs related RNA-binding proteins (RBPs) were enriched in metabolic pathways. The altered immune cell distribution, particularly macrophages and neutrophils, were perturbated by differential ASEs. The cancer hallmarks were enriched in steatotic liver grafts and closely associated with differential ASEs. Our work identified the differential ASE network with metabolic RBPs, immune cell distribution, and cancer hallmarks in steatotic liver grafts. We verified the link between steatotic liver graft injury and tumor recurrence at post-transcriptional level, offered new evidence to explore metabolism and immune responses, and provided the potential prognostic and therapeutic markers for tumor recurrence.

## 1. Introduction

Liver transplantation provides an effective treatment for selected patients with hepatocellular carcinoma (HCC), which is the second cause of cancer-mortality worldwide [1]. The extreme shortage of donor pool is still a problem, which might be alleviated by using marginal donor liver. The high incidence of non-alcoholic fatty liver disease (NAFLD) lead to the frequent application of steatotic donor liver [2,3]. However, the steatotic donor livers are vulnerable to graft injury, which may affect outcomes, including promoting late phase graft fibrosis and tumor recurrence [4,5]. The limited effective treatments to reduce liver tumor recurrence motivate the development of early diagnostic and prognostic biomarkers. The previous studies suggested that a comprehensive investigation of the transcriptome alterations is critical to understand the biological processes of HCC and to provide the potential therapeutic targets [6,7]. The applications of high-throughput RNA sequencing (RNA-seq) technology provide a powerful approach for transcriptome research. Therefore, the exploration of transcriptome alterations in steatotic graft may offer new prognostic biomarkers and potential therapeutic targets to decrease liver tumor recurrence post transplantation.

Alternative splicing is a process that removes introns and concatenate specific exons, playing important roles in post-transcriptional regulation of gene expression [8,9,10]. Various transcripts in approximately 95% multiexon human genes are produced through alternative splicing, which contribute to the diverse delivery of genetic information. Alternative splicing is critical during numerous biological processes. Abnormal alternative splicing events (ASEs) have been verified to lead to various diseases, including HCC [11,12]. The previous finding reported that alternative splicing of cell fate determinant Numb was associated with HCC and its early recurrence [7]. Another report demonstrated that differential ASEs are prevalent in HCC [12]. However, the role of ASEs on post-transcriptional modulation in steatotic liver graft after transplantation has never been explored.

The dysfunction of metabolism in liver is closely associated with the development of HCC, such as NAFLD, which has become a leading aetiology underlying many cases of HCC [13]. Our recent study demonstrated that steatotic liver graft injury was exacerbated by dysregulating mitochondrial homeostasis [14,15]. Moreover, the genes of differential ASEs were enriched in metabolism-related pathways in HCC. The expression, binding relations, and mutations of RNA-binding protein (RBP) genes largely influenced differential ASEs in HCC [12]. RBPs promote exon inclusion or skipping by binding to splicing regulatory elements. Perturbation of alternative splicing is prominently regulated by RBPs [16,17]. Multiple evidences have shown that the change in RBP levels and their activity could result in the dysregulation of alternative splicing [18,19]. Therefore, it is definitely worthwhile to investigate the association of aberrant ASEs with the dysregulation of RBPs in steatotic liver graft injury, as well as its impact on tumor recurrence.

The inflammatory responses resulting from liver graft injury reshape the immune microenvironment. Our previous findings showed that the release of inflammatory cytokines due to liver graft injury recruited immune cells, such as regulatory T cells and myeloid derived suppressor cells (MDSCs), which subsequently promoted the tumor recurrence post liver transplantation [20,21]. Furthermore, the alteration of metabolism due to liver graft injury reprogrammed immune cells. For example, the stress of liver ischemia reperfusion injury shifted the polarization of M1 to M2 macrophages, which contributed to tumor favoring microenvironment [22,23]. Nevertheless, the ASEs impacting immune cell alteration in steatotic liver graft have not been investigated.

In this study, we aimed to comprehensively analyze the ASEs in steatotic and normal liver grafts both from our clinical cohort and rat model post transplantation through RNA-seq. Our data illustrated the landscape of transcription and ASEs in liver grafts at early phase after transplantation. We also identified the differential ASEs of steatotic liver graft, together with the network of metabolic-related RBPs and the distribution of immune cells and hallmarks of cancer. The findings may not only elucidate the mechanism bridging steatotic graft injury and tumor recurrence at post-transcriptional level, but also provided potential therapeutic targets to reduce tumor recurrence. Moreover, our work about transcript and ASE based on RNA-seq in steatotic and normal liver grafts may fill in the blank of the transplant data pool and provide resources for further exploration.

## 2. Results

### 2.1. The Transcriptional Landscape of Liver Grafts in Patients and Rats at Early Stage Post Transplantation

All the raw sequencing data were qualified for the analysis. A total of 188,424 transcripts were detected to express in the liver graft transcriptome, which were categorized into five types of genes (Figure 1A, left). Although the major fraction of transcripts in pseudogenes, noncoding RNA (NcRNA) and long noncoding RNA (LncRNA) genes were annotated by Ensembl, and more than half of the protein-coding genes were newly assembled (Figure 1A, middle). These newly assembled RNAs are worthwhile to be explored for understanding the mechanism of steatotic liver graft injury and tumor recurrence. Over 50% of protein coding genes had more than five transcripts for each gene, and more than 30% LncRNA genes had more than two transcripts, while pseudogenes, NcRNA, and unannotated genes had fewer transcripts (Figure 1A, right). To investigate genomic variants in different types of assembled transcriptome, the single-nucleotide variations (SNVs) and indels were analyzed. SNVs in splicing sites occurred in all five types of genes, with higher frequency in protein coding genes (Figure 1B). Interestingly, indels in splice sites were also detected in all the gene types and more frequently found in protein coding genes (Figure 1C). Through a relatively comprehensive transcriptome analysis of human liver grafts, we found that most protein coding genes have multiple transcripts and SNVs/indels. This demonstrated the diversity of transcripts in liver grafts post transplantation.

For further exploration, the rat orthotopic liver transplantation model with normal or steatotic donors was established. The transcriptome of liver grafts was subsequently analyzed. Protein coding genes occupied almost three quarters among the 30,195 transcripts (Figure 1D, left). Consistent with clinical results, the newly assembled transcripts accounted for the majority of protein coding genes while occupying a small part in pseudogenes, NcRNA, and LncRNA genes (Figure 1D, middle). More than 30% protein coding and NcRNA genes, as well as around 10% of LncRNA, unannotated genes, and pseudogenes, had two transcripts of each gene (Figure 1D, right). The SNVs and indels were occurred in all types of genes. Moreover, the SNVs and indels in splice sites were more frequent in protein coding and unannotated genes, respectively (Figure 1E,F). Consistent with the clinical findings, the multiplicity of gene transcripts suggested that alternative splicing might play critical roles in liver graft injury of both humans and rats.

### 2.2. ASEs Were Prevalent in Both Human and Rat Liver Grafts

Alternative splicing is a major contributor to the transcriptional diversities [24]. Next, we identified the ASEs from the transcriptomic data of both human and rat liver grafts. It total, 105,572 high confidence ASEs were detected with seven types of ASEs, including A3, A5, AF, AL, MX, RI, and SE in human liver grafts. AF and SE were the most frequently observed ASEs, especially in protein coding genes (Figure 2A). The features of ASEs were further analyzed. More than half of the alternative exons were longer than 200 bps. Furthermore, the length of most RI and AL alternative exons were longer than 200 bps, while most of the SE, MX, A5, A3, and AF alternative exons were shorter than 200 bps (Figure 2B, left). In addition, the alternative exons of A3 have more intact codons (the length of intact codons is a multiple of three) than other types despite the fact that less than half of all alternative exons had intact codons (Figure 2B, right). The frequency of ASEs with varying PSI levels was further analyzed in all samples. The results demonstrated that events with high PSI (PSI = 0.8~1) levels constituted the majority of all-type ASEs. The PSI levels of ASEs in lowest (0–20%) and highest (80–100%) frequency events were obviously raised than the other middle ones (20–80%) (Figure 2C).

In rat liver grafts, 8146 high confidence ASEs were detected with seven types. Consistent with the clinical findings, AF and SE were the most frequently detected ASEs, particularly in protein coding genes (Figure 2D). More than half of the RI and AL alternative exons were longer than 200 bps, whereas most of SE, MX, AF, A3, and A5 exons were shorter than 200 bps (Figure 2E, left). Importantly, around 60% of A3 alternative exons have intact codons indicating the predominant role (Figure 2E, right). Most of the ASEs occurred in high (80–100%) frequency. In addition, the events with high PSI (PSI = 0.8~1) levels occupied the majority except for MX (PSI = 0.4–0.6) (Figure 2F). These results indicated that ASEs were prevalent and varied in liver grafts of human and rats, and they might modulate the responses to graft injury.

### 2.3. The Landscape of Differentially Expressed ASEs in Human Steatotic Liver Grafts

To explore the aberrant ASEs, the differentially spliced events between steatotic and normal liver grafts were identified. A total of seventy ASEs were detected as differentially spliced in steatotic liver grafts. All the differential ASEs were found in more than 70% patients, and most of the differential ASEs were detected in 100% of patients (Figure 3A). Furthermore, there were more upregulated differential ASEs than downregulated ones. SE accounted for the majority of upregulated differential ASEs (Figure 3B). At least 68 differential ASEs in each patient with steatotic graft were found with little variations. Moreover, the patterns of ASEs in all patients were similar, which indicated the consistency of data (Figure 3C). Multiple types of differential ASEs could be detected in the same genes (e.g., ACTB, ALDH4A1, and FBP1), while some genes harboured a single type (e.g., C1QC, CTSB, and IP6K1) (Figure 3D). These results showed that abnormities of ASEs were consistent in steatotic liver grafts of patients, despite the variations among each sample and gene.

### 2.4. The Landscape of Differential ASEs in Rat Steatotic Liver Grafts

In the rat liver transplantation model, the differential ASEs in steatotic grafts were detected in contrast with normal grafts. In total, seventy-four differential ASEs were identified in steatotic liver grafts. The majority of differential ASEs were found in more than 85% rats with few ones in over 70% rats (Figure 4A). SE and AF accounted for the majority of upregulated differential ASEs, while A5 and AF were observed in downregulated ones (Figure 4B). The number of differential ASEs in each rat was over 73, with little variations. In addition, the patterns of ASEs were consistent in each rat (Figure 4C). Consistent with the clinical results, there were various types of differential ASEs in one gene (e.g., RGD1307603 and AABR07044420.2), whereas only one type was detected in some genes (e.g., Fam210b, Trim5, and Apof) (Figure 4D). The landscape of differential ASEs in steatotic liver grafts of human and rats suggested that aberrant ASEs were induced at acute phase injury, and the subsequent inflammation might contribute to the tumor recurrence.

### 2.5. The Corresponding Genes to Common Enriched Pathways with Differential ASEs in Liver Grafts of Human and Rats Were Closely Associated with HCC Patients’ Survival

To further explore the role of differential ASEs in steatotic liver graft injury and their association with tumor recurrence, enrichment analysis was implemented on genes that harbour differential ASEs through KEGG. The genes with differential ASEs in humans were enriched in metabolism-related pathways (e.g., fatty acid degradation, glycolysis/gluconeogenesis, and arachidonic acid metabolism). Different types of ASEs were over-represented in distinctive pathways. For example, only differential SE and AL ASEs were over-represented in complement coagulation cascades (Figure 5A). In a rat model, the genes with differential ASEs were also enriched in metabolism related pathways (e.g., pyruvate metabolism, insulin secretion, and ether lipid metabolism). The various patterns of ASE types were also found in the enriched pathways. For example, only differential AF ASEs were over-represented in insulin secretion, while RI and A3 were over-represented in the intestinal immune network for IgA production (Figure 5B). Importantly, tyrosine metabolism, the metabolism of xenobiotics by cytochrome P450, and drug metabolism-cytochrome P450 were common pathways in both human and rat steatotic liver grafts.

Next, the corresponding genes were identified from the three pathways. In humans, alcohol dehydrogenase (ADH)1A, ADH1B, ADH1C, cytochrome P_450_2 (CYP2)C8, CYP2D6, CYP2E1, and homogentisate 1,2-dioxygenase (HGD) were the genes corresponding to the common pathways. Gstz1, Adh6, and RGD were corresponding genes in rats (Figure 5C). Through the gene expression profiling interactive analysis (GEPIA), the expressions of CYP2E1, ADH1A, CYP2C8, ADH1C, and HGD were significantly associated with survival of HCC patients (Figure 5D). In addition, the genes with differential ASEs in steatotic liver grafts were compared with the ones in HCC [12]. A total of 34 genes in human steatotic liver grafts coincided with HCC. Moreover, seventeen genes with differential ASEs were overlapped in rat fatty liver grafts and HCC (Appendix A). These results indicated that the perturbation of genes by alternative splicing might facilitate tumor recurrence.

### 2.6. The Splicing Regulation Was Closely Associated with Differential ASEs

The association of splicing factors with differential ASEs were further analysed. Each type of ASEs could have positive and negative correlation with one splicing related gene set. One splicing related gene set could be associated with multiple types of ASEs. The differential ASEs were strongly correlated with Post mRNA release spliceosomal complex and prespliceosome (Appendix A). These splicing factors might contribute to modulating the aberrant alternative splicing in steatotic liver grafts.

### 2.7. The Dysregulation Network of Differential ASE Related RBPs Was Enriched in Metabolism Pathways

A substantial portion of abnormal ASEs might be modulated by RBPs. Firstly, we paired the differential ASEs with corresponding RBPs to build the RBP dysregulation network. A total of 95 RBP genes were differentially expressed in human steatotic liver grafts, including 72 up-regulated and 23 down-regulated RBP genes. These RBPs were involved in many biological pathways. Importantly, the related RBPs were mainly enriched in metabolic pathways, which was consistent with our previous KEGG analysis (Figure 6A). Specially, nine metabolic related RBP genes were selected out, with six upregulated and three down-regulated ones. The metabolic pathway network of RBP genes associated with differential ASEs was shown in Figure 6B. Among them, glycolysis/gluconeogenesis was echoed with our previous KEGG pathway enrichment analysis. Our data indicated that RBPs might regulate metabolism-related pathways through ASEs, which was consistent in HCC [12]. RBPs, in the centre of the regulatory network, might play important roles in the development of liver tumor recurrence post transplantation.

### 2.8. The Severer Inflammation and Disequilibrating Distribution of Immune Cells in Steatotic Grafts Were Affected by Differential ASEs

The inflammation signature was analyzed to explore the liver graft injury in steatotic donor. The inflammation was obviously increased in steatotic grafts compared with normal ones through the enrichment analysis (Appendix A). These results verified severe steatotic liver injury. The alteration in negative regulation of cytokine production involved in inflammation response, chronic inflammatory response, positive regulation of inflammatory response, and inflammatory response were closely associated with genes involving differential ASEs, indicating that ASEs contribute to severer inflammation in steatotic grafts (Appendix A). The effects of severer inflammation in steatotic liver grafts on immune environment and tumor recurrence were further investigated.

The immuno-suppressive environment due to inflammation has contributed to the tumor recurrence as reported in our previous studies [20,21]. Through the analysis of immune scores, we found more immune cells infiltrated in steatotic liver grafts, indicating the immune environment were obviously altered due to the fatty liver graft injury. Neutrophils, macrophages, monocytes, nature killer T (NKT) cells et al. were significantly increased, while naïve CD8, Th2, and CD4 T cells et al. were significantly decreased in steatotic liver grafts (Figure 7A). The altered immune cells, such as neutrophils and macrophages, were closely associated with genes harboured differential ASEs (Figure 7B). Intriguingly, the differential ASEs of SE type were mainly positively associated with the immune cells, while almost all other types were negatively associated. The alterations in immune cell distribution due to ischemia reperfusion injury of steatotic donor liver might gradually form a tumor-favourable environment. For example, neutrophils and macrophages could differentiate into MDSCs and M2 macrophages, which could contribute to the immuno-suppressive environment to promote tumor progression [25,26]. The data suggested that the differential ASEs might modulate the immune cell distribution in steatotic liver graft injury, which might promote tumor recurrence.

### 2.9. The Differential ASEs in Steatotic Liver Grafts Were Highly Correlated with Cancer Hallmarks

To further investigate the differential ASEs, their association with cancer hallmarks was analysed. The majority of the pathways in cancer hallmarks were enriched in steatotic liver grafts, for example, glycolysis, IL-2-STAT5, inflammatory response, complement, and reactive oxygen species, which not only indicated the strong link between steatotic liver graft injury and cancer development, but also suggested the importance of both metabolism and immune response in steatotic liver graft injury (Figure 8A). Furthermore, the genes with differential ASEs were closely associated with some pathways in cancer hallmarks, such as heme metabolism, inflammatory response, complement, angiogenesis, and IL-2-STAT5 (Figure 8B). The corresponding genes to these pathways were perturbated by alternative splicing. These findings provided evidence that tumor recurrence was more frequent using steatotic donor livers. Furthermore, the alteration of gene transcripts due to ASEs in steatotic liver graft might have laid the groundwork for tumor recurrence from the beginning. The findings in this study were summarized in Figure 8C.

## 3. Discussion

The overall portrait of transcription and ASEs in steatotic liver graft both of human and rat post-transplantation were first demonstrated based on our comprehensive analysis of high-through-put RNA-seq data. In addition, we analysed the differential ASE network with metabolic RBPs, inflammation signature, immune cell distribution, and cancer hallmarks in depth. It not only provided the link between steatotic liver graft injury and tumor recurrence, but also suggested the research direction between metabolism and immune cell responses at the post-transcriptional level. The steatosis was associated with increased histological damage, hepatic function derangement, and reduced survival post liver transplantation [27]. The primary graft dysfunction was more prevalent in patients receiving donor livers with >30% fatty change [28]. Our recent study showed that graft steatosis over 10% was an independent risk factor for poor post-transplant survival and was associated with acute graft injury after living donor liver transplantation [14]. Few studies were implemented to explore the mechanism, although the phenomena of liver parenchymal abnormalities were exacerbated by steatotic liver graft injury associated with increased HCC burden, which was identified [4,29,30]. Through the analysis from the valuable biopsies in our clinical center, steatotic liver grafts with severer inflammation were obviously enriched with the genes involved in cancer hallmarks, which indicated their close relationship with tumor recurrence. Moreover, the close association of differential ASEs in steatotic liver graft with cancer hallmarks might open a new window for filling the research gap of steatotic liver graft injury and tumor recurrence.

The rat transplant model, using a steatotic donor, mimicked the clinical scenario, but it avoided patient-related confounding factors. The transcriptome analysis both in the patient and rat showed the diversity of transcripts. It was noteworthy that the substantial part of protein coding genes with transcripts were unannotated ones, which might be new players during liver graft injury. The accounts of ASEs with A3, AF, and SE type were top three in patients and consistent in the rat model. In particular, ASEs of A3 and A5 type constituted the highest fraction both in humans and rats, although the distribution of PSI value and length of ASEs were different. It will be worthwhile to further explore the specific characteristics of ASE types in liver grafts. Regarding the differential ASEs between steatotic and normal grafts, SE and AF were the most frequently observed types in human steatotic liver grafts, which was consistent with the differential ASEs in HCC patients [12]. The overlapped genes with differential ASEs of steatotic liver graft and HCC in human and rat offered important hints that these genes might contribute to tumor recurrence. Moreover, the corresponding genes in the common pathways from human and rat differential ASEs, including CYP2E1, ADH1A, CYP2C8, ADH1C, and HGD, should be noted as they were closely associated with HCC patients’ survival. The abnormality of these genes could be as prognostic markers for liver tumor recurrence at an early stage, post transplantation. Targeting these genes might provide novel therapeutic strategies to inhibit tumor recurrence.

The genes with differential ASEs were significantly enriched in metabolism-related pathways in HCC [12]. Consistently, the majority of the pathways involved differential ASEs were related to metabolism in steatotic liver grafts. Moreover, the differential ASE related RBPs were also enriched in metabolic pathways. Intriguingly, glycolysis/gluconeogenesis was found significantly associated with differential ASEs and RBPs through KEGG and RBP-ASE network analysis, respectively. The dysregulation of glycoprotein was associated with early graft injury in human liver biopsies, which was echoed by our finding [31]. Therefore, ENO1 and GAPDH, the RBPs related to glycolysis/gluconeogenesis, will be worthwhile for further study. In addition to the critical role of RBPs, the specialized splicing factors were also found to play roles on the regulation of ASEs, which may offer the hints for upstream modulation of differential ASEs in steatotic liver grafts.

The alteration of inflammation and immune cell distribution reshaped the liver graft immune environment [32]. Importantly, the increased numbers of macrophages and neutrophils in steatotic grafts were closely associated with differential ASEs in the current study. Macrophages and neutrophils played important roles in the pathogenesis of liver ischemia reperfusion injury. EP3, the prostaglandin E (PGE) receptor in monocyte-derived dendritic cells, induced IL-13-mediated switching of the macrophage phenotype from M1 to M2 in hepatic ischemia reperfusion injury [33]. Moreover, myeloid heme oxygenase 1 (HO-1) regulated macrophage polarization through favouring a M2 phenotype in liver ischemia reperfusion injury [34]. Our previous study reported that M2 promoted the development and invasiveness of HCC [26]. In addition, neutrophils could be converted into granulocytic MDSCs regulated by endoplasmic reticulum stress and lipid metabolism in cancer patients [25]. Fatty acid uptake could reprogram the neutrophils and differentiated into granulocytic-MDSCs to promote tumor progression [35]. Our recent report demonstrated that CXCL10, the increased inflammatory cytokine resulted from liver graft injury, recruited MDSCs into liver graft through TLR4/MMP9 to promote tumor recurrence post transplantation [21]. In addition, the increase in NKT and decrease in naïve CD8 T cells was also associated with differential ASEs in current study, which may contribute to the immuno-suppressive environment, facilitating tumor recurrence.

Taken together, our study not only offered the portrait of transcription and ASEs in liver grafts both from human and rat, but also identified the differential ASE network with metabolic RBPs, immune cell distribution, and cancer hallmarks in steatotic liver grafts. Our analysis verified the link between steatotic liver graft injury and tumor recurrence at the post-transcriptional level, provided new resources to explore the metabolism and immune cell responses, and offered the potential prognostic markers and therapeutics for tumor recurrence after liver transplantation.

## 4. Materials and Methods

### 4.1. Liver Graft Biopsies from Clinical Cohort

Twenty-eight patients that underwent liver transplantation in Queen Mary Hospital, The University of Hong Kong, were included in this study. Eighteen patients received steatotic donor livers (fatty change > 10% of overall macro- and micro-steatosis), while the other 10 patients received normal donors. The clinical characteristics of patients were listed in Appendix A. Over 80% of patients were Hepatitis B positive. The graft biopsies were collected at 2 h after portal vein reperfusion. Signed consent forms from each patient were acquired prior to operation. The procedures conformed to the ethical standards of the Helsinki declaration of 1975, as revised in 1983, and approved by Institutional Review Board (IRB) of The University of Hong Kong.

### 4.2. Rat Orthotopic Liver Transplantation Model

Male Sprague Dawley (SD) rats (six to eight weeks old) were obtained from the Laboratory Animal Unit, The University of Hong Kong. All animals were housed in a standard animal facility at 22 ± 2 °C under controlled 12-h light/dark cycles and had free access to chow and autoclaved water. Rats received humane care following the criteria outlined in *Guide for the Care and Use of Laboratory Animals* (*National Institutes Health publication 86–23, 1985 revision*). Experimental procedures were approved by the Committee on the Use of Live Animals in Teaching and Research, The University of Hong Kong.

The steatotic and normal donor rats were fed with 45% high-fat diet (58G8, TestDiet, Land O’Lakes, US) or regular diet for two weeks, respectively. Carbon tetrachloride (CCL4, 2 mL/kg) was injected into the recipient rats subcutaneously for four weeks to induce liver cirrhosis before the operation. The orthotopic liver transplantation model was established with small-for-size graft (ratio of graft weight to recipient liver weight was about 50%). The surgical procedure was briefly three steps: donor operation, recipient operation, and liver implantation. Generally, the survival rate was over 80%. Liver tissues were harvested at 6 h post transplantation (*n* = 3). The detailed protocols were implemented according to the previous studies [36,37].

### 4.3. RNA Sequencing

Total RNA was extracted from steatotic/normal liver grafts of patients and rat transplant model. RNA high throughput sequencing was implemented using Illumina PE150. RNA-seq data have been submitted and are available through the NCBI’s Gene Expression Omnibus (GEO GSE204919).

### 4.4. The Assembly of Transcriptome and Detection of Short Variants

Clean reads were obtained by fastp (version 0.21.0) [38]. Then, all clean reads were aligned to the human reference genome using STAR software (version 2.7.7a) with a two-step mapping strategy, which was used to utilize splice junctions from each sample. Both human and rat reference genomes were downloaded from Ensembl. The read alignments obtained from the above-mentioned two-step mapping were provided as input to StringTie (version 2.1.6) for transcriptome reassembly [39]. Annotation (human: GRCh38.103.gtf; rat: 6.0.104.gff) from Ensembl was used as the transcript model reference to guide the assembly process with the “-G” option. Firstly, transcripts were assembled individually for each sample. Then, StringTie was run in “--merge” mode to generate a set of transcripts observed in all the RNA-seq samples. The transcript level was produced by Cufflinks (version 2.2.1) in FPKM units. Transcripts with more than one exon and an expression level higher than 0.1 FPKM in at least one sample remained as high-confidence transcripts. The gene counts of each sample were calculated using RSEM (version 1.2.12) [40]. The Genome Analysis Toolkit (GATK, version 4.2.2.0) was used to call short variants with default parameters.

### 4.5. Identification of Differential ASEs and Enrichment Analysis

The splicing events were quantified by the percent spliced in (PSI) value using SUPPA (version 2.3) based on the assembled transcriptome and transcript levels [41]. PSI ranged from zero to one for quantifying seven types of AS events: skipping exon (SE), Mutually exclusive exons (MX), alternative 5′ splice-site (A5), alternative 3′ splice-site (A3), retained intron (RI), alternative first exon (AF), and alternative last exon (AL). ASEs with PSI value over 0.1 at least one sample were identified as high confidence ASEs and included in this analysis. Differential alternative splicing analyses were performed by diffSplice to calculate differential splicing between two groups. ASEs with *p* value < 0.05 were considered as differential ASEs. The parent genes of these differential ASEs were applied to Kyoto Encyclopedia of Genes and Genomes (KEGG) pathway enrichment analysis. The survival analysis of corresponding genes to common enriched pathways with differential ASEs in liver grafts of human and rats were analyzed by Gene Expression Profiling Interactive Analysis (GEPIA, http://gepia.cancer-pku.cn/ (accessed on 25 December 2021)).

### 4.6. Verification of Differentially Expressed RBP Genes and Dysregulation Network with ASEs

A catalogue of 1756 RBPs was retrieved from two previous reports [42,43]. Differential expression analysis was carried out between the steatotic and normal graft groups. Genes showing significantly differential expression (|log2(fold-change)| ≥ 1, *p* value < 0.05) were selected.

Expression of RBP genes was expected to be correlated with the PSI level of target ASEs. Therefore, we calculated the Spearman correlation for each RBP-ASE pair. RBP-ASE pairs with Spearman correlation coefficients greater than 0.5 (or less than −0.5) and a corresponding *p* value less than 0.05 were considered significantly correlated. In addition, the differential RBPs correlated with differential ASEs were analyzed to explore the related KEGG pathways. Then, a dysregulation network was built, wherein RBP-ASE pairs containing both differentially expressed RBP genes and differential ASEs were extracted.

### 4.7. Correlation Analysis of Graft Immune Cell Infiltration and ASEs

The immune cell infiltration levels were downloaded from Immune Cell Abundance Identifier (ImmucellAI, http://bioinfo.life.hust.edu.cn/ImmuCellAI#!/resource (accessed on 6 March 2022)) [44,45]. The PSI values of differential ASEs were performed correlation analyses with immune cell infiltration levels in liver grafts.

### 4.8. Gene Set Variation Analysis and Correlation with Differential ASEs

Gene set variation analysis (GSVA) was used to perform the enrichment analysis through several gene sets for splicing factor (with term ‘splice’, splicing’, ‘spliceosome’ or ‘mRNA binding’), inflammation signature, and cancer hallmark [46]. The gene sets include GENEONTOlOGY (GO) (http://geneontology.org/ (accessed on 27 December 2020)), the inflammation markers provided by He et al. [47], and cancer-related hallmarks from MigDB (http://www.gseamsigdb.org/gsea/msigdb/collections.jsp (accessed on 27 March 2022)). The PSI values of differential ASEs were performed with Spearman correlation with GSVA scores of each gene set in liver graft biopsies.

### 4.9. Statistical Analysis

Statistical analysis and data visualization statistical analyses were performed using the R software (version 4.0.1), Python (version 3.7) and Cytoscape (version 3.9.0). Data analysis and visualization tools in R software included the ggplot2, complexheatmap, GSVA, and clusterProfiler packages.

## Figures and Tables

**Figure 1 ijms-24-08216-f001:**
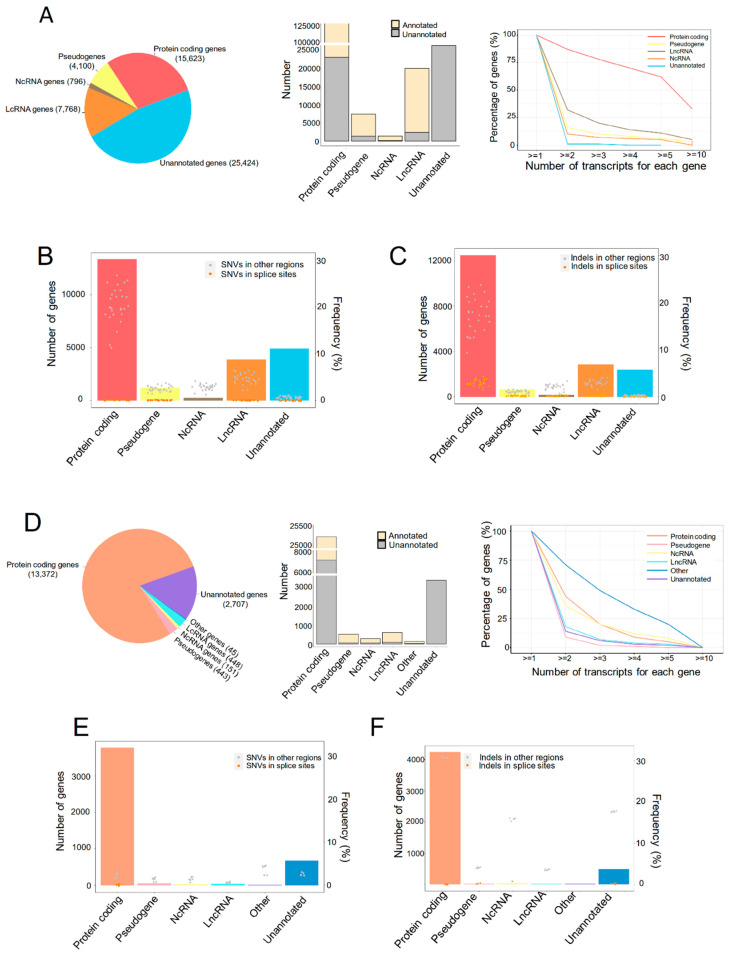
The transcriptional landscape of human and rat liver grafts. (**A**–**C**) Human liver grafts. (**A**) The landscape of transcription in human liver grafts. **Left**: the types and numbers of genes in the assembled transcriptome. **Middle**: the numbers of annotated and unannotated transcripts in different types of genes. **Right**: the distribution of transcript numbers for each gene type. (**B**) Bar plots and points showed the numbers of genes harboring SNVs for each gene type and the frequency of SNVs from each sample, respectively. (**C**) Bar plots and points demonstrated the distribution of indels and the frequency of indels from each sample for each gene type, respectively. (**D**–**F**) Rat liver grafts. (**D**) The transcriptional landscape in rat liver grafts. **Left**: the distribution of genes in the assembled transcriptome. **Middle**: the accounts of annotated and unannotated transcripts in different types of genes. **Right**: the numbers of transcripts for each gene type. (**E**) Bar plots and points indicated the accounts of genes harboring SNVs for each gene type and frequency from each sample, respectively. (**F**) Bar plots and points showed the numbers of indels for each gene type and frequency from each sample, respectively. SNV, single-nucleotide variation.

**Figure 2 ijms-24-08216-f002:**
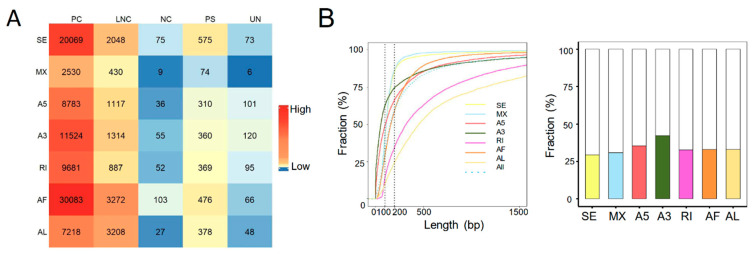
ASEs were prevalent in the transcriptome of both human and rat liver grafts. (**A**–**C**) Human liver grafts. (**A**) The distribution of ASEs with different types in each gene type of human liver grafts. (**B**) **Left**: the cumulative distribution for the length of alternative exons. **Right**: the fractions of alternative exons that have intact codons. (**C**) The events of distinct PSI levels in different frequency ranges. (**D**–**F**) Rat liver grafts. (**D**) The numbers of ASEs in different types for each gene type of rat liver grafts. (**E**) **Left**: the cumulative fraction for the length of alternative exons. **Right**: the proportions of alternative exons that have intact codons in different ASE types. (**F**) The PSI levels of ASEs in different frequency ranges. ASEs, alternative splicing events; PC, protein coding gene; LNC, long noncoding gene; NC, noncoding gene; PS, pseudogene; UN, unannotated gene.

**Figure 3 ijms-24-08216-f003:**
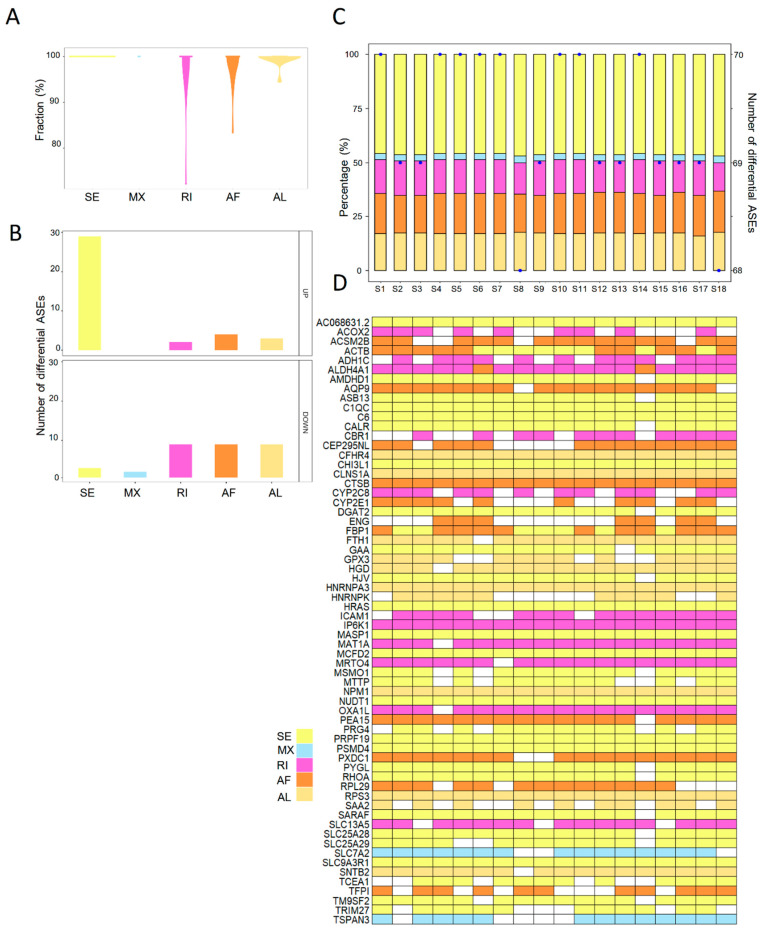
The landscape of differential ASEs in human steatotic liver grafts. (**A**) The frequency distribution of differential ASEs in different ASE types of steatotic liver grafts. (**B**) The numbers of upregulated and downregulated ASEs for each type. (**C**) Distribution of differential ASEs in the steatotic liver graft of each patient. Bars demonstrated the proportion of each ASE type. Points indicated the number of differentially ASEs in each patient. (**D**) The genes with differential ASEs showed in the heatmap. Each column represented one patient. The different ASE types for each gene in each patient were demonstrated by color cells. ASEs, alternative splicing events.

**Figure 4 ijms-24-08216-f004:**
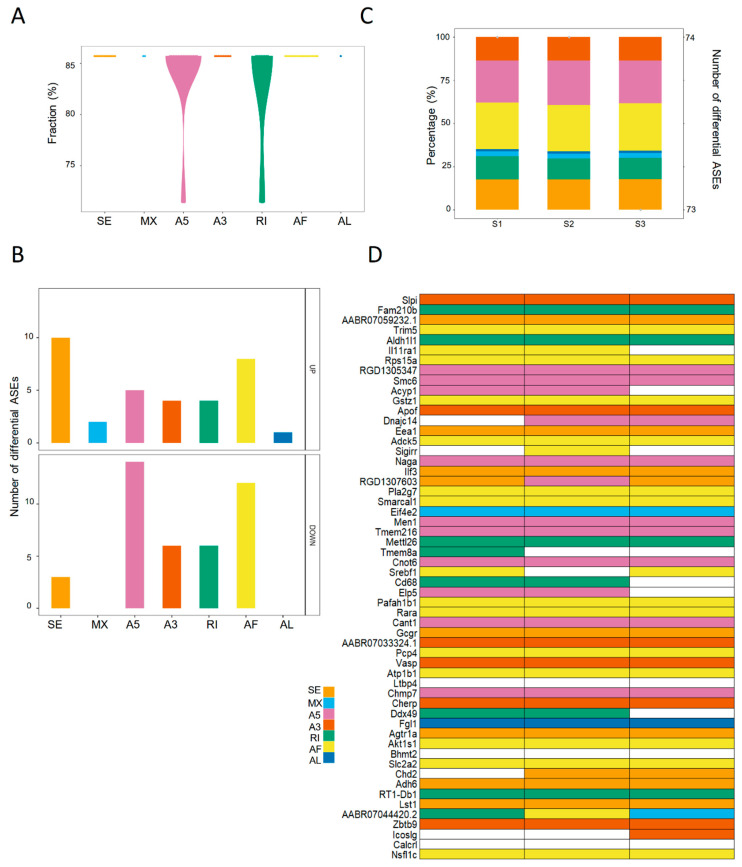
The landscape of differential ASEs in steatotic liver grafts of rat model. (**A**) The proportions of differential ASEs for each type in rat steatotic liver grafts. (**B**) The numbers of up and down-regulated ASEs for each type. (**C**) The patterns and numbers of differential ASEs in the liver graft of each rat. Bars stood for the fraction of each ASE type. Points represented the number of differential ASEs in each rat. (**D**) Heatmap demonstrated the genes with differential ASEs. Each column indicated one rat. The color cells showed the different ASE types for one distinct gene in each rat. ASEs, alternative splicing events.

**Figure 5 ijms-24-08216-f005:**
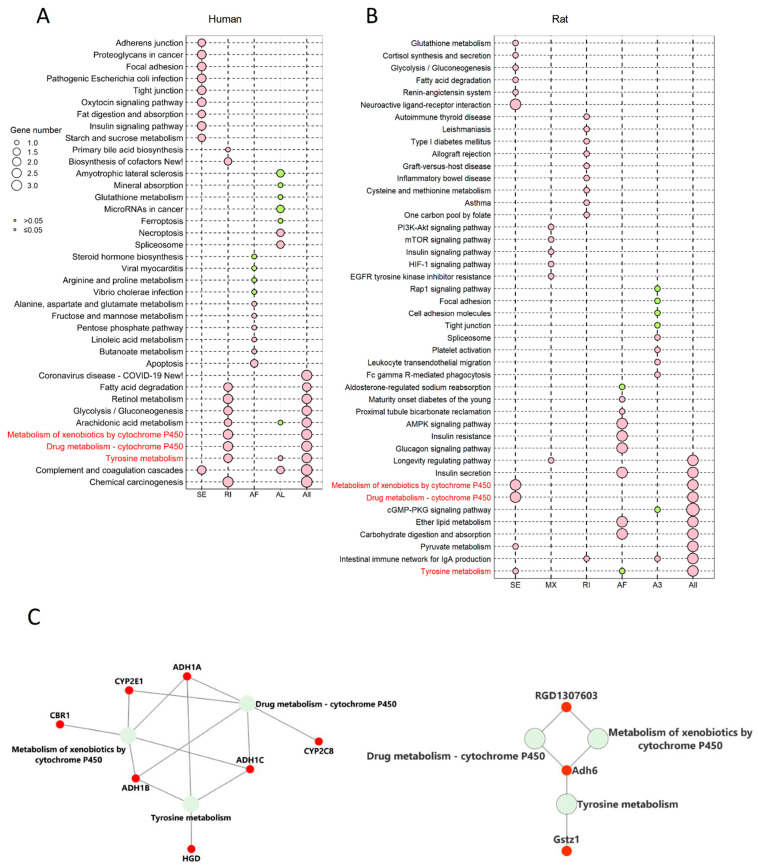
The genes corresponding to the common enriched pathways involved differential ASEs in human and rat models were highly associated with HCC survival. (**A**) KEGG enrichment analysis of genes involved differential ASEs in human steatotic liver grafts. (**B**) KEGG enrichment analysis of genes involved differential ASEs in rat steatotic liver grafts. Top 10 enriched pathways were shown. Each row represented one pathway, and each column stood for one type of ASE. The common KEGG pathways in human and rat model were highlighted with red. (**C**) The corresponding genes to the common enriched pathways involved differential ASEs. The left and right showed the genes from common KEGG pathways in humans and rats. (**D**) The human genes from the common KEGG pathways were significantly associated with the survival in HCC patients. ASEs, alternative splicing events.

**Figure 6 ijms-24-08216-f006:**
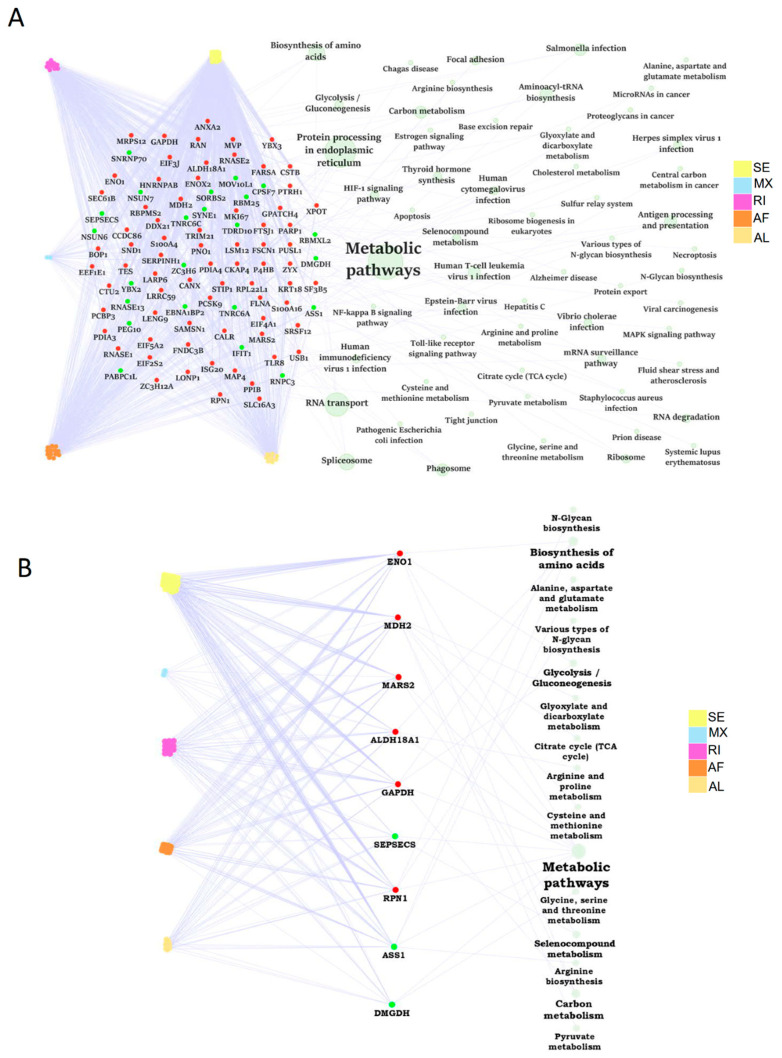
The dysregulation network of RBPs associated with differential ASEs was enriched in metabolism related pathways. (**A**) The dysregulation network of differntial ASEs, RBPs, and pathways in human liver grafts. (**B**) The regulation network of differntial ASEs and metabolic related RBPs. Labeled dots in the center indicated RBP genes. Red and green dots represented up-regulated and down-regulated RBP genes, respectively. Colored dots connected to RBP genes on the left were distinct types of differential ASEs. Green circles linked to RBP genes on the right indicated pathways. Circle size stood for the number of genes involved. ASEs, alternative splicing events; RBPs, RNA-binding proteins.

**Figure 7 ijms-24-08216-f007:**
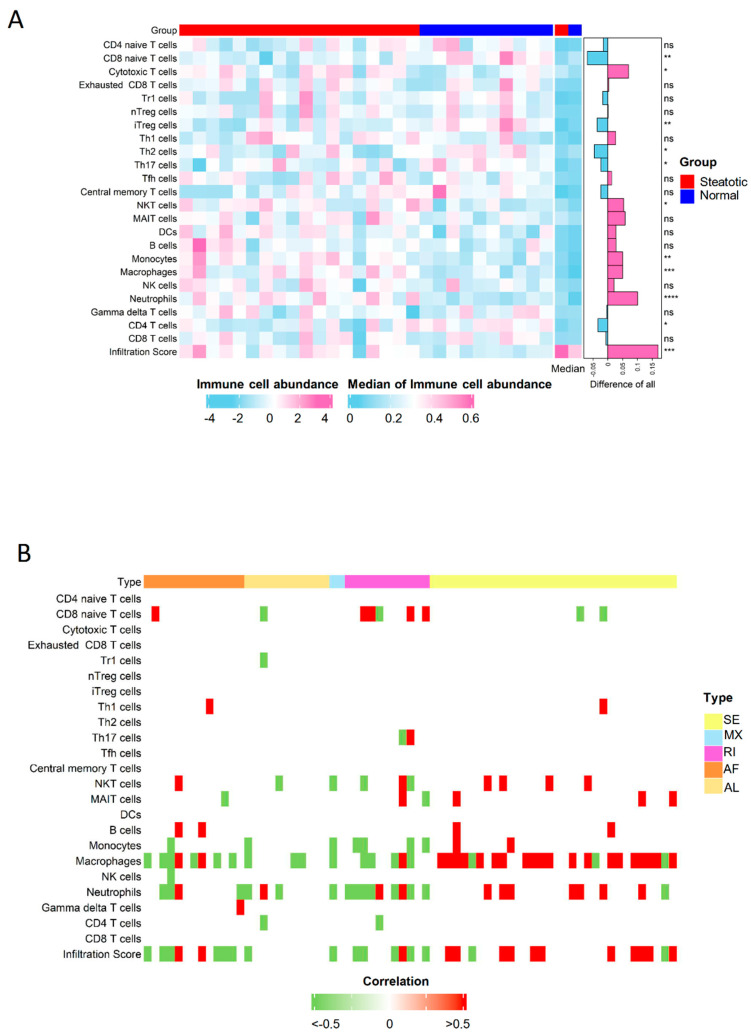
The distribution of immune cells was perturbated by differential ASEs. (**A**) The comparison of immune cell distribution between human steatotic and normal liver grafts was showed in heatmap. (**B**) The immune cells were perturbated by differential ASEs. ASEs, alternative splicing events. Degrees of statistical significance were demonstrated using standardized asterisk nomenclature (ns: no significance, * *p* < 0.05, ** *p* < 0.01, *** *p* < 0.001, **** *p* < 0.0001).

**Figure 8 ijms-24-08216-f008:**
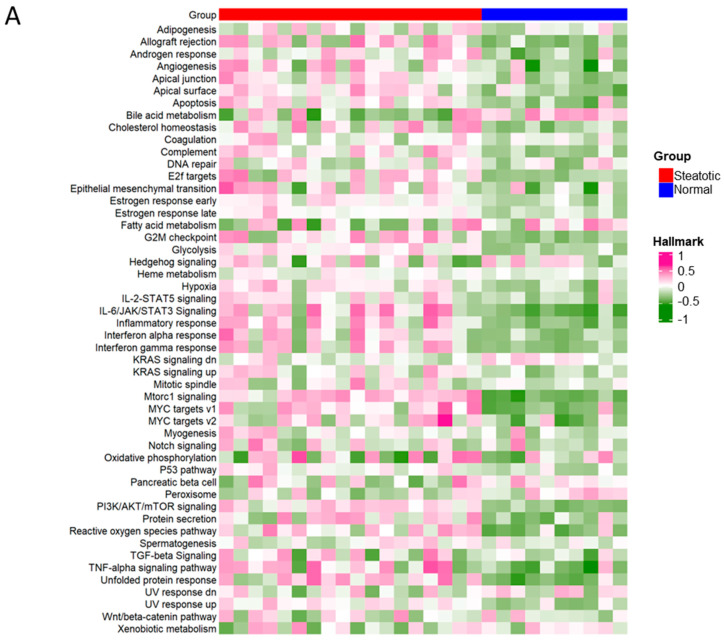
The differential ASEs in steatotic liver grafts were closely correlated with cancer hallmarks. (**A**) The cancer hallmarks were enriched in human steatotic liver grafts, which was shown in the heatmap. (**B**) The cancer hallmarks were affected by differential ASEs. (**C**) Research summary: Our work depicted the landscape of transcripts and ASEs in liver grafts of humans and rats, and it identified the differential ASEs in steatotic liver grafts. We picked out the genes harbouring differential ASEs (from common pathway in human and rat) associated with HCC patients’ survival and together with the network of metabolic-related RBPs, distribution of immune cells (neutrophils and macrophages), and hallmarks of cancer.

## Data Availability

RNA-seq data have been submitted and are available through the NCBI’s Gene Expression Omnibus (GEO GSE204919).

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
