# Peer review of "The Landscape of Aberrant Alternative Splicing Events in Steatotic Liver Graft Post Transplantation via Transcriptome-Wide Analysis"

_ijms, 2023, doi:10.3390/ijms24098216_

Round 1

Reviewer 1 Report

The authors presented a very interesting article. Among the comments are the following. I would like to see a more detailed description of the liver transplantation technique and the results on the survival rate of the donor liver fragment. I understand that the article is mostly bioinformatics, but usually transcriptome analysis data is confirmed by real-time PCR.

Author Response

Dear Reviewer,

Thanks for the valuable comments and suggestions. Our point-to-point responses to the comments are outlined in the attachment. We appreciate this opportunity to submit a revised manuscript and hope the editors and reviewers will find it sufficiently improved to justify publication in International Journal of Molecular Sciences.

Nancy Kwan Man (on behalf of the authors)

Professor

Department of Surgery

Li Ka Shing Faculty of Medicine, School of Clinical Medicine

The University of Hong Kong

Tel: +852 39179646

Reviewer 2 Report

This manuscript would be an important contribution to the field. Data is very well organized and the results are sufficiently discussed. I did find some grammatical errors and typos which authors should correct before submission. Overall its a well organized study and I don't have any specific concerns.

Author Response

(The authors gave the same response as above.)

Reviewer 3 Report

The manuscript "The landscape of aberrant alternative splicing events in steatotic liver graft post transplantation via transcriptome-wide analysis" is well written and presents valuable insights into the molecular mechanisms of steatotic liver graft injury and tumor recurrence after transplantation. The authors used high-throughput RNA-seq to identify differential alternative splicing events (ASEs) in both human and rat transplant models, and found specific genes, including CY2E1, ADH1A, CY2C8, ADH1C, and HGD, that were significantly associated with HCC patients' survival. Additionally, the authors observed perturbations in immune cell distribution, particularly macrophages and neutrophils, and enrichment of cancer hallmarks in steatotic liver grafts that were closely associated with differential ASEs.

However, there are a few minor revisions that the authors should consider. Firstly, they could provide more information on the potential prognostic and therapeutic implications of their findings. This would help readers better understand the significance of the study's results and their potential impact on clinical practice. Secondly, the authors should ensure that the format of the legend for Figure 5 is consistent with that of the other figures. This would improve the readability and overall presentation of the manuscript.

Author Response

(The authors gave the same response as above.)

Reviewer 4 Report

The you for the opportunity to review the submission “The landscape of aberrant alternative splicing events in steatotic liver graft post transplantation via transcriptome-wide analysis” by Lui et al.  The authors are building upon the data showing that elevated graft steatosis predisposes to graft injury and fibrosis.  This, in turn, creates an environment which could be more favorable for tumor recurrence after liver transplantation. The authors identify prior roles for abnormal alternative splicing in HCC, abnormal ASE in metabolic pathways in NAFLD, and their intersection in RNA-binding protein genes. Using RNA-seq in clinical specimens and a rat orthotopic liver transplant model, the authors identify differentially regulated ASE, aberrant RBP networks, and immune cell signatures which are proposed to promote rapid HCC recurrence post-transplant.

General Comments.

Most of the manuscript data is obtained from the post-transplant human tissue biopsies.  In these specimens, the authors identify aberrant ASE in the steatosis cohort, drill down to specific genes of interest, utilize pathway analysis to identify RBP networks, and show correlations between these findings and the post-transplant immune environment.  The manuscript does not utilize the rat data for quantitative mechanistic analysis.  The rat data would be important for a methods and validation focused manuscript, but the overall flow of the manuscript would be substantially improved if the rat data was excluded. While the data is very interesting, the analysis and discussion are speculative, overstating the link between the observations and clinical outcomes.

Specific Comments.

In the authors citations, it is important to note that the studies linking HCC recurrence (or graft loss) to steatosis underestimate well-characterized factors associated with HCC recurrence risk.  For instance, in reference #4, the manuscript is focused on graft loss, not HCC graft loss due to recurrence, and the steatosis cohort HCC population is extremely small and >50% outside of Milan criteria.  This is probably not the best citation to infer “steatosis-driven graft injury…. may promote tumor recurrence”.

The patient cohort is predominantly HBV-associated HCC, which is disclosed in the supplement. If the manuscript were more primarily focused on HCC, this would be critical to disclose in the body of manuscript.

In the authors characterization of steatosis, are they referring to macrosteatosis or the overall contribution of macro- and micro-steatosis?  The assessment and methods for quantifying graft steatosis should be outlined in the methods.

The ultimate purpose and relevance of the rat model to the overall manuscript is unclear. The correlation between the models according to Figure 1 is overstated. Further the rat data ultimately obscures the flow of the manuscript.

The GEPIA methods and analysis in Figure 5 are not outlined in the methods section, making it difficult to interpret the data. It appears the genes with differential ASEs between donor tissue from steatosis versus without steatosis were also associated with overall survival in HCC, and ultimately favored higher expression of each gene target versus lower expression.  The authors conclude this section by stating that the genes perturbed by alternative splicing in the transplant tissue might facilitate tumor recurrence post-transplant.  There are several issues with this extrapolation. There is no clear directionality to the overall RNA or protein expression levels for the target genes between steatosis and without steatosis.  Second, the assumption that gene expression in the transplant tissue would directly influence circulating tumor cells is tenuous and unexplored.  Third, if the rationale is that this expression profile mimics HCC and may promote de novo HCC, the current data regarding de novo HCC versus HCC recurrence in early-onset (1-2yr) HCC post-transplant significantly favors circulating tumor cells as the source of HCC recurrence.  The relationship between post-transplant tissue gene expression and its correlation with expression in primary HCC tissue, although tenuous, should be developed in the discussion.

In the following excerpt, “Furthermore, the genes with differential ASEs were closely associated with some pathways in cancer hallmarks, such as Heme metabolism, Inflammatory response, Complement, Angiogenesis, IL-2-STAT5 (Fig. 8B). The corresponding genes to these pathways were perturbated by alternative splicing. These findings provided the evidence that tumor recurrence was more frequent using steatotic donor liver.”, it is unclear how any relationship between these pathways was associated with an observed higher incidence of HCC recurrence. No HCC recurrence data is presented in the manuscript.

Overall.

Overall, the manuscript and findings are very interesting, but are very qualitative with significant extrapolation/speculation.  The rat model data does serve the manuscript well and obscures the overall focus of the manuscript while cluttering the early portion of the results section.   There is extensive use of the “might” and “may” which then become evidence of a “novel link” in the discussion. Although the data is very interesting and creates a foundation for ongoing research, the results and discussion should be revised to concisely present the results and avoid overly extending the results in the discussion section.

Author Response

(The authors gave the same response as above.)

Round 2

Reviewer 4 Report

The authors have addressed my concerns.